# Chromosome Distribution of Highly Conserved Tandemly Arranged Repetitive DNAs in the Siberian Sturgeon (*Acipenser baerii*)

**DOI:** 10.3390/genes11111375

**Published:** 2020-11-20

**Authors:** Larisa S. Biltueva, Dmitry Yu. Prokopov, Svetlana A. Romanenko, Elena A. Interesova, Manfred Schartl, Vladimir A. Trifonov

**Affiliations:** 1Institute of Molecular and Cellular Biology SB RAS, Lavrentiev Ave., 8/2, 630090 Novosibirsk, Russia; bilar@mcb.nsc.ru (L.S.B.); rosa@mcb.nsc.ru (S.A.R.); vlad@mcb.nsc.ru (V.A.T.); 2Department of Ichthyology and Hydrobiology, Tomsk State University, Lenin Ave, 36, 634050 Tomsk, Russia; e.interesova@ngs.ru; 3Developmental Biochemistry, University of Wuerzburg, Biocenter, Am Hubland, 97074 Wuerzburg, Germany; phch1@biozentrum.uni-wuerzburg.de; 4Xiphophorus Genetic Stock Center, Texas State University, 601 University Drive, 419 Centennial Hall, San Marcos, TX 78666-4616, USA; 5Novosibirsk State University, Novosibirsk, Pirogova, 2, 630090 Novosibirsk, Russia

**Keywords:** *Acipenser baerii*, sturgeon karyotype, whole-genome duplication, paralogs, polyploidy, acipenserid minisatellite, satellite DNA, tandem repeats

## Abstract

Polyploid genomes present a challenge for cytogenetic and genomic studies, due to the high number of similar size chromosomes and the simultaneous presence of hardly distinguishable paralogous elements. The karyotype of the Siberian sturgeon (*Acipenser baerii*) contains around 250 chromosomes and is remarkable for the presence of paralogs from two rounds of whole-genome duplications (WGD). In this study, we applied the sterlet-derived acipenserid satDNA-based whole chromosome-specific probes to analyze the Siberian sturgeon karyotype. We demonstrate that the last genome duplication event in the Siberian sturgeon was accompanied by the simultaneous expansion of several repetitive DNA families. Some of the repetitive probes serve as good cytogenetic markers distinguishing paralogous chromosomes and detecting ancestral syntenic regions, which underwent fusions and fissions. The tendency of minisatellite specificity for chromosome size groups previously observed in the sterlet genome is also visible in the Siberian sturgeon. We provide an initial physical chromosome map of the Siberian sturgeon genome supported by molecular markers. The application of these data will facilitate genomic studies in other recent polyploid sturgeon species.

## 1. Introduction

The order Acipenseriformes represents one of the most basal lineages of vertebrates and has a long evolutionary history with fossil records dating back to 200 million years ago. The order includes around 27 extant species from two families (Acipenseridae and Polyodontidae). Phylogenetic relationships among the acipeserids are not fully resolved. Molecular and morphological studies produced controversial results [1,2]. In addition to the 1R and 2R whole-genome duplication (WGD) events, which are common for all vertebrates, sturgeon genomes have undergone a series of additional WGDs. It has been hypothesized from cytogenetic and molecular genetic data, and from genome size estimates that the first round of an Acipenseriformes-specific WGD independently occurred both in the ancestors of Acipenseridae (Ac1R), and Polyodontidae (Po1R) [1,3]. Among the acipenserids, several additional WGD events took place, giving rise to ≈250-chromosome and ≈370-chromosome species [4]. However, in most cases, it remains unclear whether these are auto- or allopolyploidization events. These processes are still ongoing in different sturgeon species where spontaneous polyploids have been recorded [5]. Previously it was believed that rediploidization following WDGs seems to be close to completion in species after Ac1R and is still actively going on in species after Ac2R [1]. However, genomic studies have revealed that in sterlet (*Acipenser ruthenus*) the gene number and gene expression level are much closer to the tetraploid status [6]. At the same time, multiple rearrangements, specifically involving middle-size chromosomes, have accompanied karyotype evolution [7,8].

Unusual karyological characteristics of this ancient vertebrate group include the presence of dot-like microchromosomes, which—as was first suggested by Ohno et al. [9]—represents an ancestral state and unites them with other basal ray-finned fishes, the Holostei, coelacanth, lungfish, and some tetrapods (reptiles and birds). Later it was shown that the majority of chromosomes previously detected as “dot-like” microchromosomes morphologically can be classified as meta/submetacentrics or acrocentrics [10]. Another karyological peculiarity of this group is the absence of strictly fixed chromosome numbers that are described to vary between post-Ac1R species from 99 to 120, and between post-Ac2R species from 239 to 270 (for review see [4]). There is a considerable variation within each species, as well as intraindividual variations that complicate the identification of the precise modal chromosome number. The variations cannot always be explained as artifacts of chromosome preparation [11,12]. Previous assumptions about the potential presence of a variable number of heterochromatic supernumerary chromosomes [13] have not been confirmed by applying of microdissection-derived small chromosome-specific probes [7,8]. In some acipenserid species, the association between major and minor rDNA gene clusters was detected, which were located close together on the same chromosomes [14]. This occurs quite infrequently in mammals, but was also observed in some lower eukaryotes [15] and in some fish species [16,17,18]. Sturgeon genomes are also characterized by substantial intraindividual variation in the sequence of the 18S rRNA gene, which is a quite rare phenomenon in animal genomes [19]. The genome of sterlet has recently been sequenced and assembled [6], and thus, it can serve as a reference genome for further studies of acipenserid cytogenetics and genomics.

Tandem repeats quickly diverge in evolution. They are often species-specific, and less frequently, they are common or highly similar for a taxon of a higher rank, for instance, the cetacean-specific satDNA [20]. Two satellite DNA repeats have been characterized in acipenserids: The expansion of the PstI repeat in the sturgeon lineage is dated shortly after their divergence from the Polyodontidae around 200 Mya [21], while the HindIII satellite obviously arose in a clade of sturgeons about 170 Mya, as this repeated DNA family is absent in the early-diverged North Atlantic clade [21,22]. The stability of repetitive unit length and sequence of both these ancient repeats (HindIII and PstI) indicates an extremely low rate of molecular evolution [21,22]. It is notable that the number of chromosomes carrying the HindIII repeat family increased tenfold after additional rounds of polyploidization in several acipenserid clades [23], while other repeats, such as PstI and rDNA, did not expand so radically [14,23,24]. The process of polyploidization can be accompanied by the expansion of transposons, as shown in salmonids [25], but this has not been observed after teleost specific 3R WGD [26]. Previously, we detected a bimodality in the distribution of repeats in the sterlet karyotype. The chromosomal distribution of the repetitive DNA fraction (C_o_t30) revealed differences between small and large chromosomes [27]. Satellites with a short repetitive unit expanded mainly across multiple small chromosomes, while repetitive DNAs with a long repetitive unit are biased to have chromosome-specific distribution and some of them distinguish paralogous chromosomes [28]. 

In earlier studies, we revealed paralogous chromosomes derived from the duplication of the 60-chromosome ancestor, as well as post-Ac1R chromosomal rearrangements [6,7,27]. Sequencing of DNA from isolated sterlet chromosomes and alignment to the spotted gar reference genome provided information on gene content, and syntenic blocks track their evolutionary path after the polyploidization events. It was postulated that after Ac1R, the sterlet karyotype underwent multiple interchromosomal rearrangements, but different chromosomes were involved in this process unequally [6,7,8]. 

To study the sturgeon karyotype after the second round of polyploidization (Ac2R), we chose as a model the Siberian sturgeon (*Acipenser baerii*), which together with sterlet belongs to the Atlantic clade of acipenserids. As the karyotypes of 250-chromosomal species are hard to characterize cytogenetically, the application of previously elaborated molecular markers helps to start an initial standardization of multi-chromosomal sturgeon karyotypes. To describe subchromosomal genomic changes caused by expansion or loss of tandemly arranged DNA repeats in sturgeon species with a high polyploidy level, we applied the previously obtained set of sterlet-derived repetitive DNAs and chromosome-specific probes as physical markers for FISH on Siberian sturgeon chromosomes. We found that the Siberian sturgeon genome contains all repetitive DNAs described in sterlet, but their genomic distribution is different from what would be expected from a simple whole-genome doubling.

## 2. Materials and Methods 

### 2.1. Ethics Statement

All applicable international, national, and/or institutional guidelines for the care and use of animals were followed. All experiments were approved by the Ethics Committee on Animal and Human Research of the Institute of Molecular and Cellular Biology, Siberian Branch of the Russian Academy of Sciences (IMCB, SB RAS), Russia (order No. 32 of 5 May 2017). This article does not contain any studies with human participants performed by any of the authors.

### 2.2. Sample Origin

The adult Siberian sturgeon (*A. baerii*) female (originated from the Ob basin population) was obtained from the Tomsk fish farm (“SibEco”). The sterlet (*A. ruthenus*) repetitive DNA probes (*Arut19A, ArutF26A, Arut30A, Arut40A, Arut57A, ArutF167A, Arut219A, Arut434A,* and *Arut802A)* and Cy3-labeled oligonucleotides ((AC)_15_, (AAG)_9_, (ACAG)_7_, (ACAT)_8_, (AAC)_10_, and (AAT)_8_) were reported earlier [28]. The ribosomal and HindIII satellite DNA probes used here were presented previously by Romanenko et al. [27]. The set of microdissected DNA probes of sterlet chromosomes, including ARUT 1p and ARUT 2p (probe N5); ARUT 3 (R70) and ARUT 4 (R61), ARUT 6 (R68), ARUT 7 (R69), ARUT 8 (R64), ARUT 14 (R58), and ARUT 57 (R3) were described earlier [8]. The primers for U2 small nuclear RNA genes have been described in [29].

### 2.3. Chromosome Preparation, Staining, and Painting

An optimized protocol for cell culture and chromosome preparation for sturgeons was established previously [27]. Here we used tissue cultures obtained from the swim bladder. GTG (G-banding by trypsin using Giemsa) differential staining was performed according to Seabright [30], and CBG (C-bands by barium hydroxide using Giemsa) banding was done according to Sumner [31]. Fluorescence in situ hybridization (FISH) using repetitive and microdissection-derived DNA probes was performed as described previously [27]. To clarify the chromosome morphology, we also used an inverted DAPI karyotype of the same metaphase. To identify the chromosome of origin, as well as paralogous regions, we carried out dual-color FISH with different probes in a series of pairwise experiments and compared the sizes, morphology, and banding pattern of labeled chromosomes. Images were captured using VideoTest-FISH 2.0 software (Imicrotec) with a JenOptic charge-coupled device (CCD) camera (Jena, Germany) mounted on an Olympus BX53 microscope (Shinjuku, Japan). Images were processed using Paint Shop Photo Pro X3 (Corel, Ottawa, ON, Canada).

## 3. Results

### 3.1. Nomenclature of A. baerii Chromosomes

Since *Acipenser baerii* chromosomes are supposed to be derived from the genome duplication of a 120-chromosomal ancestor, we suggest here to use previously characterized and sequenced sterlet chromosomes as a reference for *A. baerii* chromosome description. We suggest referring to Ac2R paralogs of the Siberian sturgeon as “ABAE”. For example, while ARUT 1 and ARUT 2 are Ac1R paralogs, ABAE 1 and ABAE 2, derived from the ancestral element homologous to ARUT 1, will be Ac2R paralogs. We also suggest distinguishing the derived elements of related genomes from the paralogs of other orders depending on the degree of proximity. For example, ABAE 1 and ABAE 2 are closely related to ARUT 1, while ABAE 3 and ABAE 4 paralogs are more closely related to ARUT 2, which is paralogous to ARUT 1. Thus, ABAE 1 and 2 are inparalogs, while the pairs (ABAE 1 and 2) and (ABAE 3 and 4) may be regarded as outparalogs (Figure 1). ARUT 1 will be ortholog of ABAE1 and ABAE2, but interspecific outparalog to ABAE3 and ABAE4.

### 3.2. Conventional Cytogenetics of the Siberian Sturgeon Karyotype: GTG- and CBG-Banding

#### 3.2.1. GTG-Banding of *A. baerii* Chromosomes

Karyotyping was performed by analyzing 21 metaphases. The number of chromosomal elements ranged from 238 to 252, with the majority of cells (16) ranging from 240 to 246. Thus, the chromosome number of the ABAE 1f specimen is 245 ± 7. This is close to the chromosome number of specimens from the Lena River (2n = 249 ± 5) [32] and from the fish factory in Italy (2n = 246 ± 8) [33]. GTG-banded chromosomes were paired and arranged in rows according to their size and morphology (Figure 2). 

Almost half of the chromosomes are small, poorly distinguishable elements. Figure 2 shows a karyotype from a metaphase with 248 chromosomes (100 bi-armed chromosome pairs and 24 acrocentric chromosome pairs). 

#### 3.2.2. C-Banding of *A. baerii* Chromosomes

C-blocks were identified in the pericentromeric regions of all acrocentrics and the majority of small metacentrics (Figure 3). The large and middle bi-armed chromosome pairs lack heterochromatin blocks except for a few middle-size chromosomes with visible C-blocks in the pericentromeric regions. On some small chromosomes double blocks of heterochromatin were detected on both p- and q-arms, and about 20 pairs seemed fully heterochromatic. Figure 3 shows a metaphase with 252 chromosomes after CBG-staining.

### 3.3. Chromosomal Mapping of Microsatellite DNAs 

Chromosomal mapping of five microsatellite sequences, (AC)_n_, (AAG)_n_, (ACAG)_n_, (ACAT)_n_, and (AAC)_n_, revealed numerous signals on small chromosomes of *A. baerii*. Around 80–100 signals were detected mainly in subtelomeric regions, with about 10 chromosomes producing particularly strong signals, often at both (p- and q-) subtelomeric blocks (Figure 4a and Appendix A). A different pattern of subchromosomal localization was revealed on NOR-bearing chromosomes, where all microsatellite sequences except (AC)_n_ were located in the pericentromeric regions. Notably, most of the microsatellite DNA probes showed signals in pericentromeric regions of four middle-sized metacentrics (Figure 4a).

### 3.4. Localization of Arut19A, Arut30A, Arut40A, and Arut57A Satellite DNA Probes on Chromosomes of A. baerii

Satellites *Arut19A, Arut30A, Arut40A,* and *Arut57A* mapped to multiple sites of the Siberian sturgeon small chromosomes. Their individual distribution on chromosomes was different. *Arut40A* and *Arut57A* accumulated preferably on terminal ends of the q-arms, *Arut19A* had mostly a pericentromeric location, while *Arut30A* was located more at the ends of p-arms. The number of signals per metaphase varied from 12–14 for *Arut19A* (Appendix A) to around 80 for *Arut30A* (Appendix A), *Arut40A* (Figure 4b), and *Arut57A* (Figure 4c). In addition, there were several small chromosome pairs with strong signals at the ends of both arms, sometimes spreading over the whole arm. These satellites, except *Arut19A,* also localized in the pericentromeric regions of the NOR-bearing chromosomes (Figure 4c). 

*ArutF167A* accumulated in more than 40 acrocentrics of different sizes, located on both p-and q- arm pericentromeric blocks (Figure 5a). Dual-color FISH of *ArutF167A* with whole chromosome probe derived from ARUT 57 labeled pericentromeric regions of four small chromosomes with different morphology in *A. baerii*: Two submetacentrics (ABAE 86) and two subtelocentrics (ABAE 100), whose q-arms were painted by the probe ARUT 57 (Figure 5b). 

### 3.5. Satellites with Chromosome-Specific Location

In *A. baerii*, the repeat *Arut434A* was localized on 20 chromosome pairs: ABAE 1–11, 13, 15, 16, 23, 101–104, and 108. Dual-color FISH of the *Arut434A* in combination with whole chromosome painting probes of the sterlet chromosomes ARUT 1–7, 9, and 14 reveal orthologs in the *A. baerii* (Figure 6, Figure 7 and Appendix A). As expected, each of these probes hybridized with two *A. baerii* chromosomes, identifying them as putative inparalogs, while only one chromosome of ARUT 4 and 7 orthologs were conserved. The rest orthologs of ARUT 4 and 7 underwent fissions and fusion after Ac2R with the formation of acrocentric and submetacentric chromosome pairs each. The intensity of the *Arut434A* signal differed between Siberian sturgeon outparalogs; the signal was clearly observed on chromosomes ABAE 1 and ABAE 2, putative orthologs of sterlet chromosome ARUT 1, but poorly detected on ABAE 3 and ABAE 4, putative orthologs of ARUT 2 (Figure 6a). 

Similarly, the differences in the intensity of *Arut434A* on both arms of sterlet chromosomes ARUT 3 and ARUT 4 were also conserved for their Siberian sturgeon orthologs ABAE 5–7, 101, and 102 (Figure 6b and Appendix A). Moreover, one of the orthologs of ARUT 4 has undergone fission leading to the formation of 2 pairs of large acrocentrics: ABAE 101 and 102. This rearrangement was accompanied by the accumulation of *Arut434A* on ABAE 102 and the reduction of copy number of this repeat on ABAE 101.

*Arut434A* is conserved across ARUT 5 and its orthologs ABAE 8 and 9, but significantly reduced on ARUT 6 and its orthologs ABAE 10 and 11 (Appendix A).

One of the *A. baerii* orthologs of ARUT 7 has undergone significant changes, resulting in the formation of the submetacentric chromosome ABAE 23 and acrocentric chromosome ABAE 108 (Figure 5c). According to dual-color FISH with probes of the ARUT 7 whole chromosome-specific paint and ARUT 7q-arm paint, it can be proposed that the fission that led to the formation of two acrocentrics was followed by a fusion of one acrocentric with a chromosome/chromosome arm resulting in the biarmed ABAE 23, while the second acrocentric is present as ABAE 108. The repeat *Arut434A* marks these chromosomes weakly, as well as the orthologs of ARUT 14 (ABAE 103 and 104). In comparison to ABAE 23 and ABAE 108, ABAE 103 and 104 have different sites of *Arut434A* localization, retaining the pattern characteristic of ARUT 14.

The distribution of *Arut434A* on the Siberian sturgeon orthologs of ARUT 8 corresponds to that in the sterlet: The repeat was not detected on inparalogs ABAE 12 and 14. The opposed situation was observed in the orthologs of ARUT 9, the signal was localized more brightly in one of the orthologs (ABAE 15) and absent in its inparalog (ABAE 16), although sterlet chromosome ARUT 9 showed multiple sites of *Arut434A* (Appendix A). 

We expanded this work by localizing the conserved U2 small nuclear RNA gene cluster, which is restricted to sterlet paralogs ARUT 10 and 12 (Appendix A). It was localized in the pericentromeric regions of two inparalog pairs: ABAE 19 and 20; and 25 and 35, respectively (Figure 6d).

Localization of *Arut434A* and small nuclear RNA gene cluster U2 on chromosomes of sterlet and respective orthologs of the Siberian sturgeon are schematically shown in Figure 7.

### 3.6. NOR-Bearing Chromosome-Specific Repeats

Satellites *Arut802A, ArutF26A,* and microsatellite (AAT)_n_ were localized on NOR-bearing chromosomes of the Siberian sturgeon. Hybridization signals for satellites *Arut802A, ArutF26A,* and microsatellite (AAT)_n_ were restricted to two of the three pairs of NOR-bearing chromosomes of the sterlet, while ribosomal probe 18S/28S rDNA was localized on all three chromosome pairs: ARUT 30, 31, and 32 [27]. In the Siberian sturgeon, the ribosomal probe 18S/28S rDNA marked seven pairs of NOR-bearing chromosomes (instead of the expected six pairs, which should be present after the WGD). Four large 18S/28S rDNA sites were detected on the inparalogs ABAE 30 and 50, which are orthologs of the sterlet chromosome 30 (Figure 8a and Figure 9). ABAE 30 is bigger than ARUT 30, and probably had undergone a fusion with a chromosome or chromosome arm after the Ac2R. The ABAE 50 is heteromorphic chromosome pair: One homolog presented as a submetacentric chromosome similar in size and morphology to ARUT 30 but the second homolog as a small chromosome with a reduced p-arm (Figure 2 and Figure 9). The remaining 10 signals overlap with signals from satellite *Arut802A* on two pairs of paralogs: ABAE 51 and 56, 52 and 55, and additionally on ABAE 63. Remarkably, only one of the *A. baerii* paralogs from each pair (ABAE 51 and 52) is similar to its sterlet orthologs (ARUT 30 and 31), while the other paralogs (ABAE 56 and 55) appear smaller in size. Detection of an additional signal on chromosome ABAE 63, which probably resulted from rearrangements of the NOR-bearing chromosomes after Ac2R, was completely unexpected.

Repeats *ArutF26A* and (AAT)_n_ colocalized on seven pairs of small chromosomes (ABAE 51, 52, 56, 64, 68, 76, and 81), whereby signals on three (ABAE 51, 52, and 56) of these pairs overlapped with *Arut802A*. Inparalogs ABAE 68 and 76, as well as ABAE 64 and 80, were different from each other by size, but had similar banding patterns (Appendix A, Figure 8b and Figure 9). 

The satellite *Arut219A* colocalized with 5S rDNA, revealing four strong signals on two pairs of small submetacentric chromosomes (ABAE 91 and 93) that are morphologically different from their ortholog ARUT 41 (Figure 8c). Besides, satellites *Arut19A, Arut40A* (Figure 4b), and *Arut57A* were also detected on these chromosomes. The most unusual observation was the colocalization of satellites *Arut802A* and 5S rDNA on chromosome ABAE 91 (Appendix A). A plausible scenario of NOR-bearing chromosome changes after the Ac2R is schematically shown in Figure 9.

Thus, using the chromosome-specific repeats isolated from sterlet and the U2 small nuclear RNA gene probe, as well as chromosome-specific painting probes, it was possible to differentiate the orthologs of the 12 largest sterlet chromosomes (ARUT 1–10, ARUT 12, and ARUT 14), four small NOR-bearing chromosomes (ARUT 30–32, and ARUT 41) and the tiny chromosome ARUT 57 in the genome of the Siberian sturgeon (Appendix A). All chromosome-specific repeats were mapped on the Siberian sturgeon chromosomes (Appendix A).

## 4. Discussion 

Acipenseriformes is one of the relict vertebrate groups with a long evolutionary history, whose representatives went through one or several rounds of WGD. Despite the fact that phylogenetic relationships between the species are not fully resolved, there is an obvious correlation between the level of ploidy of the species and its taxonomic position (Table 1). All species from the taxa Polyodontidae and *Scaphirhynchus*, as well as the North Atlantic or sea sturgeon lineage, including *A. sturio/A. oxyrinchus* have 2n ≈ 120 chromosomes, which resulted from a single round of WGD. The early-diverging groups of the Atlantic clade also have 2n ≈ 120, while the rest of this clade, as well as all species of the Pacific clade, have undergone additional rounds of WGD and have 2n ≈ 250 or even higher chromosome numbers, e.g., 2n ≈ 370 in *A. brevirostrum* [4]. As a result, karyotypes of most sturgeon species consist of multiple chromosomal elements, the exact number of which is mostly uncertain either, due to real variation between individuals or cells or imprecise cytological methods. Doubling of the number of known cytogenetic markers (i.e., Ag-NOR, 18S/28S rDNA, and 5S rDNA) is usually expected for species that underwent the Ac2R WGD [3,34]. However, the distribution of molecular markers of Acipenseridae-specific repetitive DNA families (such as Pst1 and HindIII satellites) suggests a more complex picture. After the last round of polyploidization, the number of HindIII satellite-containing chromosomes significantly increased from 2–8 to 49–80 [23,24], while the number of PstI satellite-carrying chromosomes increased not so dramatically from 4 to 12 [3]. It should be noted that both repeats are taxon-specific for Acipenseridae: The Pst1 satellite is typical for all species, whereas the HindIII satellite appeared after separation of the early-diverged North Atlantic lineage [21,34]. The presence of these repeats in *H. huso* is another evidence that the taxon *Huso* (including also *H. dauricus* from the Pacific Clade) is paraphyletic and based solely on morphological characters [21,34].

The phylogeny of Acipenseridae is still debated, but the most common view is that *A. baerii* and *A. ruthenus* belong to different branches of the Atlantic clade [34,38]. In comparison to the sterlet karyotype, where most of the chromosomes are bi-armed except two acrocentric pairs [27], the number of acrocentrics significantly increased in the Siberian sturgeon and reached almost fifty. The majority of *A. baerii* chromosomes previously detected as microchromosomes [32] corresponded in our analysis to meta/submetacentrics or acrocentrics. Discrepancies in the ratio of biarmed and acrocentric chromosomes between earlier studies and our work are explained by improved karyotypic resolution. Additional rounds of polyploidization in acipenserid lineages did not simply lead to a doubled set of ancestral chromosomes as one would expect from comparing the chromosomal numbers but also changed the structure of their karyotypes. Thus, the post-Ac2R karyotypic changes are apparent by a massive increase in the number of acrocentrics for the *A. baerii*.

The distribution of heterochromatin blocks on *A. baerii* chromosomes is similar to that in sterlet: All acrocentrics and small chromosomes are heterochromatin-rich, while the large chromosomes are heterochromatin-poor, except for the large acrocentrics. A comprehensive analysis of the main fractions of sterlet tandemly repeated DNAs conducted earlier gave us knowledge of the composition of heterochromatin [28]. Despite the long-lasting process of divergence between both species, sterlet-specific repeats were conserved in the Siberian sturgeon genome, identifying their orthologs. We showed that the chromosomal distribution of microsatellites is restricted mainly to telomeric and sometimes to pericentromeric regions of small chromosomes in both species, and their number is roughly doubled in *A. baerii*. The major fractions of satellite DNAs can be divided into three groups according to their distribution on chromosomes of the sterlet and Siberian sturgeon, which sometimes overlap: Satellite DNAs with multiple location sites mostly on small chromosomes; satellite DNAs with chromosome-specific localization mainly on large and medium-sized chromosomes, and satellite DNAs specifically located in NOR-bearing chromosomes. 

In comparison to sterlet, the number of signals of the first group repeats (*Arut30A, Arut40A,* and *Arut57A*) on the Siberian sturgeon chromosomes approximately doubled as expected after Ac2R. The *Arut19A* repeat was exceptional, as it painted a similar number of regions in both species. This result might indicate a certain reduction in the repeat copy number after the Ac2R and requires further research. The satellite *ArutF167A* belonging to the HindIII repetitive DNA family dramatically expanded in the Siberian sturgeon karyotype. In sterlet, *ArutF167A* showed a chromosome-specific localization with strong signals in the pericentromeric region of two chromosomes: Acrocentric chromosome 14 and the tiny chromosome 60 [28]. In *A. baerii*, the number of signals increased to 40, and most of them were detected on acrocentric chromosomes, including orthologs of ARUT 14 and 60. The chromosome distribution of this repeat correlates with changes in the structure of the karyotype after the last WGD event by the nonlinear increase in the number of acrocentrics. It is possible that either the generation of new acrocentrics was triggered by the accumulation of these repeats or vice versa that the formation of novel acrocentrics expands of this specific repetitive DNA family. These data indicate that changes in karyotype structure after the Ac2R WGD were accompanied by amplification of some repetitive elements. Noteworthy that colocalization with ARUT chromosome 57 specific probe demonstrated that one inparalog (ABAE 100) had similar morphology to ARUT 57; and it seems to be conserved in toto after Ac2R, while the other inparalog (ABAE 86q) underwent a fusion (with ABAE 86p). In contrast, orthologs of ARUT 14, inparalogs ABAE 103 and 104 do not have significant differences in morphology and size (Table 2). 

The localization of chromosome-specific repeats and probes made it apparent that post-WGD evolution of the Siberian sturgeon involved many complex inter- and intrachromosomal rearrangements. The distribution of satellite *Arut434A* on sterlet chromosomes provides a possibility to distinguish paralogs resulted from Ac1R, due to a difference in signal intensity [28], which could not be discriminated using the current set of ARUT chromosome microdissection probes [7,8]. Surprisingly, using satellite *Arut434A* as a probe, we could clearly distinguish the Ac1R-outparalogs of sterlet chromosomes in the Siberian sturgeon genome according to the intensity of the signals on the sterlet paralogs and their orthologs (Figure 7 and Appendix A). Colocalization of the *Arut434A* repeat with chromosomal probes of large sterlet chromosomes (ARUT 1–9 and 14) traces rearrangements of the first five chromosomes of the putative acipenserid ancestor (AcA) in the genomes of *A. ruthenus* and *A. baerii* (Table 2). The orthologs of three of them (AcA 1, 3, and 5) remained unchanged in both karyotypes, while the orthologs of chromosome AcA 2 underwent a centromeric break in the *A. baerii* karyotype. The most rearranged were the orthologs of chromosome AcA 4, which underwent a series of consecutive centromeric fissions and fusions in the karyotypes of both species.

Simultaneous detection of ribosomal probes 18S/28S rDNA and 5S rDNA with repeats specifically located in NOR-bearing chromosomes (*Arut802A, ArutF26A,* (AAT)_n_, and *Arut219A*) revealed complex rearrangements of small chromosomes (Figure 9, Table 2). Unlike large chromosomes, all five small ancestral chromosomes (AcA 15, 16, 20, 21, and 29) have a complex rearrangement in the sturgeon karyotype, which include not only fissions and fusions, but also transformations that led to the appearance of heteromorphic homologs (ABAE 50) and the emergence of ribosomal sites on an additional chromosome (ABAE 63) that does not seem to have inparalogs in the sturgeon genome.

Using the obtained data, we tried to reconstruct an approximate scenario of chromosomal reorganizations of 11 ancestral chromosomes in the genomes of *A. ruthenus* and *A. baerii*. Generally, we observed a trend in the rate of rearrangements of small chromosomes, which was much higher than of the large elements of the genome. A similar trend was found in the sterlet genome earlier [8]. The small ancestral chromosomes have undergone numerous transformations, including fissions and fusions, sometimes leading to the appearance of heteromorphic homologs and the emergence of a new NOR-bearing chromosome. Perhaps, it is these unusual karyotype phenomena that lead to a lack of certainty in the chromosome numbers of sturgeon karyotypes.

The evolution of the sterlet genome leading to partial functional diploidization and genome reduction occurs at different rates in various parts of the genome [7], but the rate of re-diploidization is extremely low in comparison to other polyploid species [6]. High conservation of some repetitive elements between sturgeon genomes, due to the slow molecular evolution of the taxon in general efficiently uses a set of sterlet repetitive DNA, as well as microdissection-derived chromosome-specific probes in sturgeon genomes of higher ploidy. As a perspective, the universal cytogenetic markers for sterlet and Siberian sturgeon, described in this article, could be applicable for the whole Acipenseridae family, and may serve as anchors for future genome assemblies. 

We constructed a detailed chromosomal map of Siberian sturgeon and used a set of sterlet-specific repeats and chromosomal probes to show complex genome rearrangements after Ac2R. These included changes in the chromosomal structure of the karyotype, chromosome fusions and fissions, repeat expansions and reductions, as well as turnover of ribosomal DNA clusters. Using the obtained data, we reconstructed the evolutionary reorganizations of 11 ancestral chromosomes after Ac1R and Ac2R.

## Figures and Tables

**Figure 1 genes-11-01375-f001:**
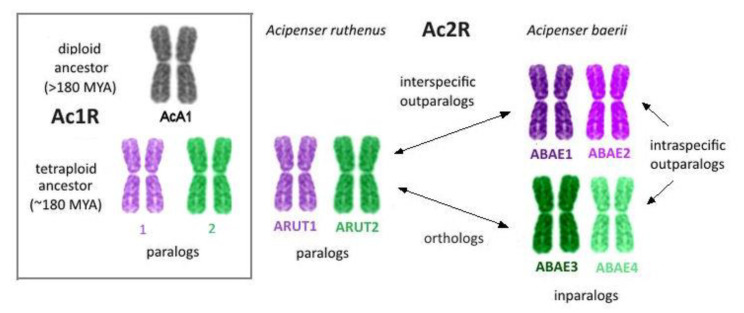
Different types of chromosome homology relations in polyploid acipenserids, which genomes underwent either one (Ac1R), or two (Ac2R) whole-genome duplication events.

**Figure 2 genes-11-01375-f002:**
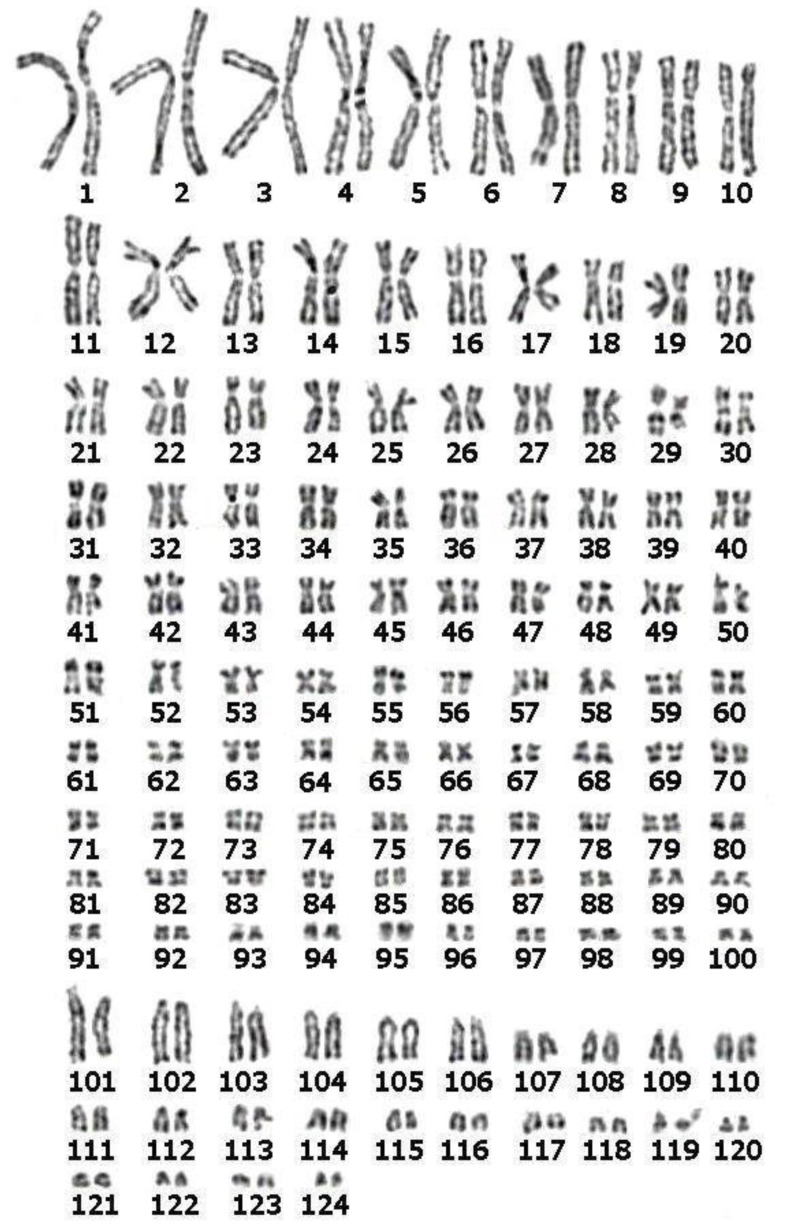
GTG-banded karyotype of the Siberian sturgeon (*Acipenser baerii*) (2n = 248). The morphologies of some homologs in pairs differ in this image (for example, pair 10). Observed heteromorphism was not confirmed by analyzing other metaphases and can be explained by either chromosome spreading artifact or by a rare chromosome rearrangement.

**Figure 3 genes-11-01375-f003:**
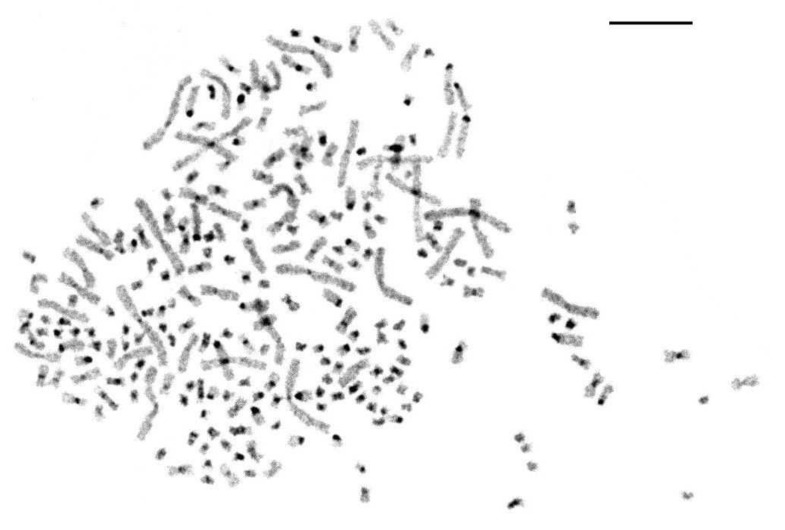
CBG-stained chromosomes of the Siberian sturgeon. The scale bar is 10 µm.

**Figure 4 genes-11-01375-f004:**
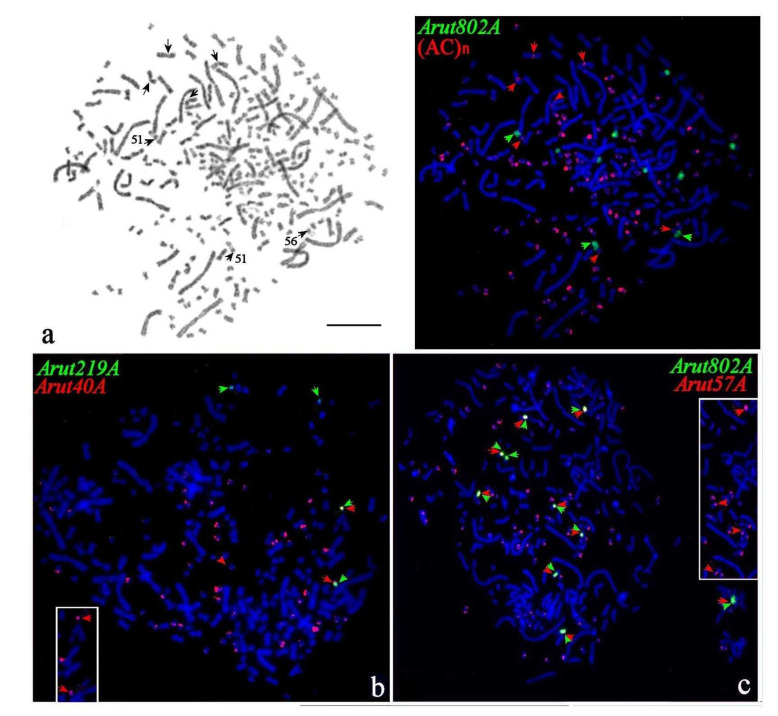
FISH of repetitive DNA probes on metaphase plates of the Siberian sturgeon (**a–d**). (**a**) Satellites (AC)_n_ (red) and *Arut802A* (green) on GTG-banded metaphase plate (**left**). Arrows mark four middle-size metacentric chromosomes and NOR-bearing chromosomes with dual signals; (**b**) satellites *Arut40A* (red) and *Arut219A* (green), arrows mark chromosomes with dual signals; (**c**) satellites *Arut57A* (red) and *Arut802A* (green), arrows mark NOR-bearing chromosomes with dual signals, in the right insert the part of the plate is represented only with *Arut57A.* The scale bar is 10 µm.

**Figure 5 genes-11-01375-f005:**
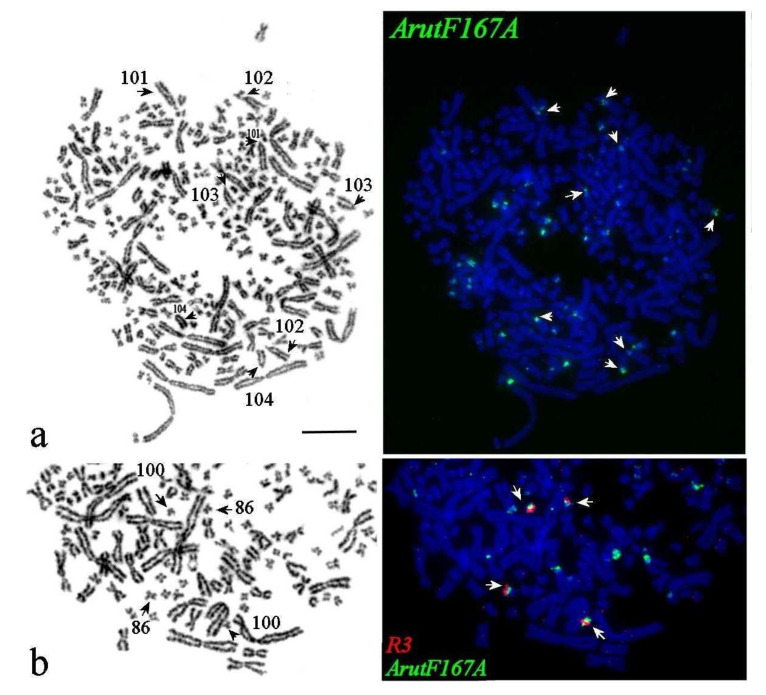
FISH of repetitive DNA probes on metaphase plates of the Siberian sturgeon (**a**,**b**). (**a**) ArutF167A on GTG-banded metaphase plate (**left**), arrows mark big acrocentric chromosomes with signals and arrowheads strong signals on ABAE 104. (**b**) satellite ArutF167A (green) and painting probe ARUT 57 (R3, red) on the part of GTG-banded metaphase plate (**below**). Arrows mark ABAE chromosomes 86 and 100. The scale bar is 10 µm.

**Figure 6 genes-11-01375-f006:**
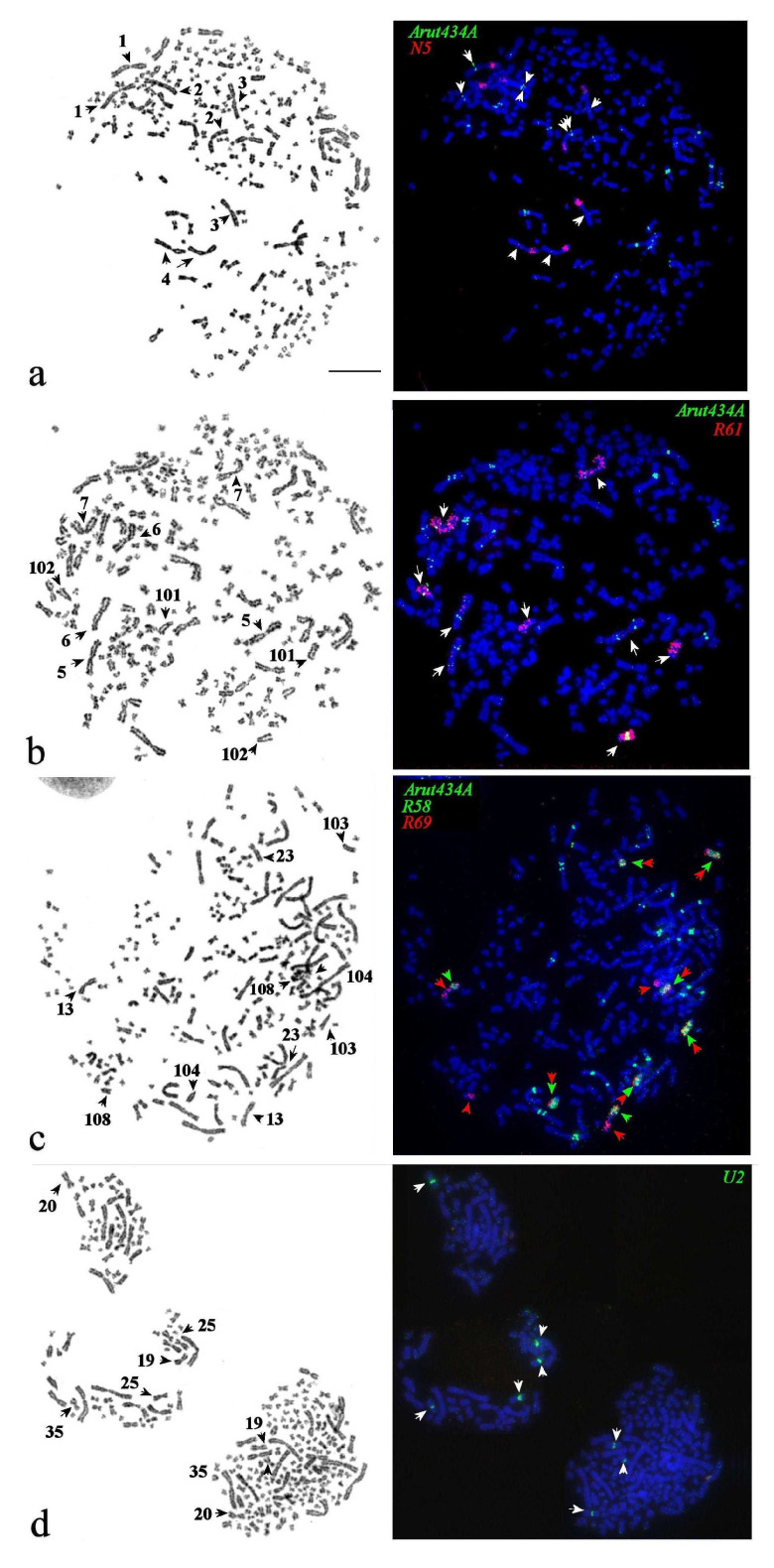
FISH of repetitive and microdissection-derived painting probes (**right**) on GTG-banded metaphase plates (**left**) of the Siberian sturgeon (**a**–**d**). (**a**) Arut434A (green) and microdissection-derived painting probe N5 (red) for ARUT 1p and 2p colocalized on ABAE 1–4; (**b**) Arut434A (green) and microdissection-derived painting probe R61 (red) for ARUT 4 colocalized on ABAE 7, 101, and 102, sometimes very weakly marked ABAE 5 and 6, orthologs of ARUT 3; (**c**) Arut434A (green) and microdissection-derived painting probes R69 (red) for orthologs of the ARUT 7 (ABAE 13, 23, and 108) and ARUT 14 (ABAE 103 and 104), and R58 (green) for orthologs of ARUT 7q (ABAE13 and 23)and ARUT 14. (**d**) U2 (green) localized into chromosomes 19, 20, 25, and 35. Arrows marked chromosomes with signals. The scale bar is 10 µm.

**Figure 7 genes-11-01375-f007:**
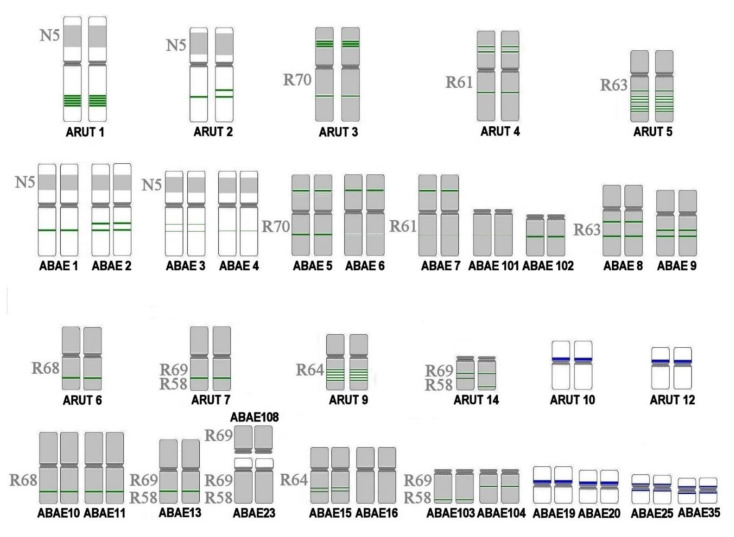
Schematic distribution of the repeat Arut434A (green), small nuclear RNA gene cluster U2 (blue), and chromosome-specific probes (grey) on the orthologs of 11 sterlet chromosomes in the Siberian sturgeon karyotype.

**Figure 8 genes-11-01375-f008:**
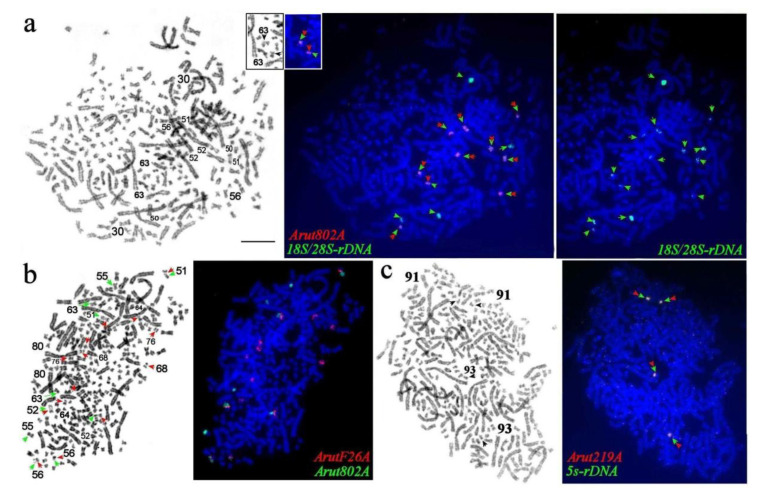
FISH of repetitive and microdissection-derived painting probes (**right**) on GTG-banded metaphase plates (**left**) of the Siberian sturgeon (**a**–**c**). (**a**) 18S/28S rDNA (green) and *Arut802A* (red); (**b**) *Arut802A* (green) and *ArutF26A* (red); (**c**) 5S rDNA (green) and *Arut219A* (red). Chromosomes with signals are marked and numbered; double arrows indicate chromosomes with colocalized probes. The scale bar is 10 µm.

**Figure 9 genes-11-01375-f009:**
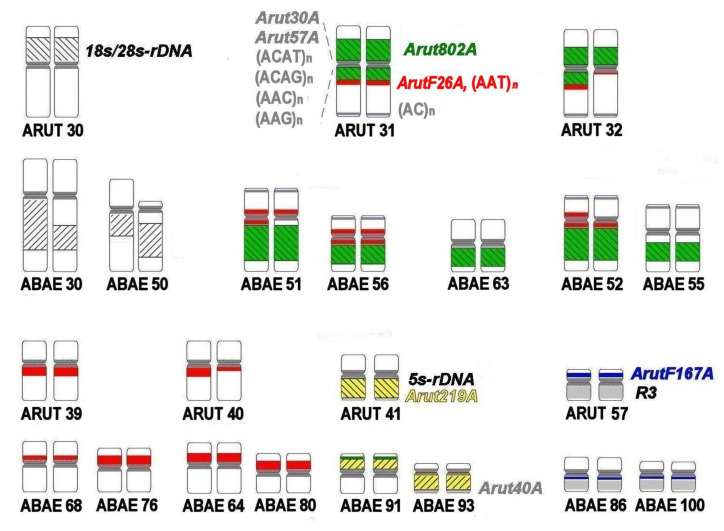
Scheme of tandemly arranged repetitive DNA dynamics in the karyotypes of sterlet (ARUT) and Siberian sturgeon (ABAE), based on the chromosomal localizations of sterlet-specific repetitive DNAs and microdissection-derived painting probes on Siberian sturgeon chromosomes. The names of repetitive and chromosome-specific probes are given near the chromosomes of their origin.

**Table 1 genes-11-01375-t001:** Molecular and cytogenetic characteristics of Acipenseriformes species.

Species	Ploidy Level	Chromosome Number	Number of Chromosomes Carrying Molecular Markers, or Their Presence (+)/Absence (−) in the Genome					Reference
			Ag-NOR	18S/28S rDNA	5S rDNA	HindIII satDNA	PstI satDNA	
*Polyodon spathula,*	2n	120	4	6	2	-	-	[11,22]
*Scaphirhynchus* *Platorynchus*	2n	112						[1]
*A. sturio* ^1^	2n	121 ± 3	6	8	2	-	+	[22,23,24]
*A. oxyrinchus* ^1^	2n	121 ± 3				-	+	[22,24]
*Huso huso* ^2^	2n	116 ± 4	4	6	2	8	+	[1,22,23]
*A. stellatus* ^2^	2n	118 ± 2	6	6-8		10	4	[1,22]
*A. ruthenus* ^2^	2n	118 ± 2	4	6	2	2(8)	+	[1,8,22,23,35]
*A. fulvescens* ^2^	4n	262 ± 6					+	[22,33]
*A. brevirostrum* ^2^	6n	372					+	[10,22]
*A. baerii* ^2^	4n	249 ± 5		10–12	4	38 ± 3	+	[22,23], this study
*A. naccarii* ^2^	4n	239 ± 7		10–12	4	50 ± 4	+	[22,23,36]
*A. gueldenstaedtii* ^2^	4n	250 ± 8			4	80 ± 4	12	[22,23,36]
*A. sinensis* ^3^	4n	264 ± 4					+	[22,37]
*A. transmontanus* ^3^	4n	248 ± 8		10–12		60	+	[13,22,23]

^1^ Sea Sturgeon Clade; ^2^ Atlantic Clade; ^3^ Pacific Clade

**Table 2 genes-11-01375-t002:** Correspondence between some chromosomes of the proposed ancestor (AcA), *Acipenser ruthenus*, and *Acipenser baerii* and putative chromosomal reorganizations in both karyotypes.

NN	Chromosome Numbers			Chromosome Rearrangements
	AcA, 2n ~ 60	ARUT, 2n ~ 120	ABAE, 2n ~ 250	
1	1	1	1; 2	
		2	3; 4	
2	2	3	5; 6	centromeric fission of one ortholog of ARUT 4 resulted in two acrocentrics: ABAE 101 and 102
		4	7; 101;102	
3	3	5	8; 9	
		6	10; 11	
4	4	7	13; 23;108	centromeric fission of one ortholog of the putative ancestral chromosome 4 formed two acrocentrics: ARUT 7q and 14; fusion of ARUT 7q and 7p; centromeric fission of one ortholog of ARUT 7 with formation of 2 acrocentrics: ABAE 23q and 108; fusion of ABAE 23q and 23p
		14	103; 104	
5	5	8	12; 14	
		9	15; 16	
6	6	10cent	19p cent	
			20p cent	
		12cent	25pq cent	
			35pq cent	
7	15	30	30; 50	fusion of ABAE 30q and 30p; loss of ABAE 50p in one of the homologues
8	16	31	51; 56	fissions of ABAE 55 and 56; formation of a new site of 18S/28S rDNA on ABAE 63
			63	
		32		
			52; 55	
9	20	39	68; 76	fusion in orthologs ARUT 39 and 40, resulting in the appearance of ABAE 64 and 68 which are bigger than their inparalogs ABAE 80 and 76, respectively
		40	64; 80	
10	21	41	91; 93	fission of one of ARUT 41 ortholog resulted in the appearance of small submetacentric ABAE 93, which is almost twice smaller than its inparalog ABAE 91
11	29	57	86q; 100	fusion of ABAE 86q and 86p

cent—centromeric region.

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
