# Peer review of "Chromosome Distribution of Highly Conserved Tandemly Arranged Repetitive DNAs in the Siberian Sturgeon (Acipenser baerii)"

_genes, 2020, doi:10.3390/genes11111375_

Round 1

Reviewer 1 Report

This manuscript represents an interesting and useful contribution to phylogenesis of the ancient taxon of the sturgeons which went troughout one or several round of WGD. In particular the numerous probes used, highlighted an extremely complex and fragmented situation  not previously imaginable, revealing changes in the genome due to polyploidization events. Molecular cytogenetics demonstrates in this manuscript  its great potential and the possibility of providing information that is not possible to have from the only molecular phylogenetics. I think that cytogenetics can provide useful information for a multidisciplinary approach to the study of speciation and hybridization processes even when molecular biology can not do it. In this manuscript, the methods are properly used and the figures are of excellent quality. I think that this manuscript is of high level and that absolutely it should be  accepted for publication in Genes with an English revision and some minor revisions that I listed below:

Figures: they are very beautiful, but often too small and some details are indistinguishable. Adding some inserts might help to understand. Moreover often the white arrows are too small.

Line 67-68- there are a lot of papers on 45S and 5S colocalization in fish and I think they need to be mentioned

Line 90    60-chromosome ancestor

Line 140   add commas: ABAE2, derived from the ancestral element homologous to ARUT 1, will be Ac2R paralogs

Line 179     Figure 3

Line 198    a)  there are different types of large and small arrows, what do they indicate? Dual signal are not visible. could they fit into an insert?

Line 206 delete a period

Figure 5 letters in bold. This figure is difficult to understand some inserts could help. In a, b and d arrows are too small

Line 283-284      this repeated sentence should be deleted

Line 2912-293  if this is the most unusual observation, this should be shown in a figure (maybe in an insert?)  in the main text

Table 1 and Table 2 shoud be reformatted and compacted but they are very interesting

Line 359     one??? I don’t understand

Line 368  it is better to delete the comma after noteworthy

Line 383   this fact should be better underlined in the legend for Figure 5

Line 391 It should be better explained the term co-localization clarifying if it means close location in the same chromosomal region or interspersion between repeated sequences. As far I can understand from the figures, both these situations are present in A. baerii karyotype (Figures 4, 5, 7)

Author Response

Dear Reviewer,
Thank you for reviewing our manuscript and helpful comments on our paper, which we now have incorporated and which greatly improved the manuscript. We have carefully considered all suggestions and made the respective changes throughout the manuscript.

Best regards,

Dmitry Prokopov

Reviewer 2 Report

The paper by Biltueva et al is a cytogenetic characterization of the genome of the Siberian sturgeon. Authors have used satDNA-based probes to understand the evolutionary history of chromosomes in this species  whose genome is believed to result from two rounds of whole genome duplications (WGD).

The paper is very well written, schemes, figures and tables have been chosen with great care and clearly make the understanding much easier than in courant cytognetic papers. This work represents a nice example of how repeat-targeted probes can be used to infer chromosome rearrangements that occur during evolution and also can be used as cytogenetic markers.

Author Response

Dear Reviewer,
Thank you for reviewing our manuscript. We have carefully considered all suggestions from all reviewers and made the respective changes throughout the manuscript.

Best regards,

Dmitry Prokopov

Reviewer 3 Report

Acipenseriformes is among the most basal lineages of vertebrates. In addition to the 1R and 2R whole genome duplications (WGD), they have undergone further WGDs, leading to plyploid condition. In present article, the Siberian sturgeon Acipenser baerii is chosen as a model to analyze and describe subchromosomal genomic changes occurred during the second round of polyploidization.

The combination of elevate number of chromosomes per metaphase and occurrence of numerous very small-sized elements make cytogenetic studies in Acipenseriformes species challenging. In this context, the choice to integrate conventional and molecular cytogenetics is certainly a valuable approach, holding potential to understand the complexity of Acipenseriformes genome evolution.      

Overall, I enjoyed reading the article that is clearly written, up-to-date, and well documented. In my opinion, the authors undertook the study with appreciable scientific rigor and methodological accuracy.

I only have a few minor comments for your consideration (below).

Introduction:

  • Line 67-68 “but was also observed in some lower eukaryotes” association between major and minor rDNA genes has been observed and documented by molecular cytogenetics in several fish species (e.g. Amorim et al., 2016, 10.3897/CompCytogen.v10i4.10227; Caetano de Barros et al., 2017, DOI 10.1007/s10750-015-2583-8; Pisano and Ghigliotti, 2009, doi:10.1016/j.margen.2009.03.006; just to mention some), I suggest to include this information.

Materials and methods:

  • Please provide information on the microscope and camera here
  • Lines 129-130 “To clarify the chromosome morphology, we also used an inverted DAPI karyotype of the same metaphase” are the images occasionally shown in the figures (e.g. Figure 4a – 4d – 4e)? Not clear to me.

Results:

  • Line 168 “Figure 2 shows …after CBG-staining” this is not what is reported in the legend to figure. Please amend.
  • Figure 4, in my opinion there are too many images here. I suggest to split the figure into two figures, the first including images a, b, and c, the second expanding d and e (so that also image e would be as large as the others). In my opinion this may aid clarity.

Typographical errors:

  • Line 132, missing full stop
  • Line 206, delete a full stop
  • Line 415, change “Sibirean” to “Siberian”  

Author Response

(The authors gave the same response as above.)

Reviewer 4 Report

The proposed manuscript (MS) is focused on karyotype evolution and genome reorganization of Acipenseridae after WGD events and it shows some peculiar and intriguing phenomena in nature. I have two main points/questions: (i) Surprising finding is that chromosome numbers vary within one individual. High chromosome number as it is in Acipenser baerii might be affected by cytogenetic error and there is a question: What is the extent of karyotypic error against intra-individual variability of chromosome numbers (between cells within one individual). (ii) What is an origin of polyploidy of individuals/species with higher ploidy levels within Acipenseridae. Comments and suggestions more in details bellow:

1) Line 61: Authors state intraindividual variations in the term of chromosome numbers. Lines 153-4: Authors determined chromosome number of ABAE 153 1f specimen is 245±7. If I understand correctly, specimen ABAE 153 has cells with various chromosome numbers within one body. It would be very rare in nature and it seems as a cell mosaicism, does not it? Inter-individual variability was already published e.g., in a diploid-polyploid complex of the fish genus Carassius (Kalous and Knytl, 2011). Can authors support the claiming about intra-individual variability of chromosome numbers by citations within Acipenseriformes or within whichever fish order? Are there any statistical analyzes supporting intra-individual variability of chromosome numbers against cytogenetic errors? If yes please cite them.

2) Interesting question is if it is known that A. baerii arose via autopolyploidization or allopolyploidization, eventually what are parental species. Could authors add this answer on the section of introduction or discussion?

3) Starting on the line 124: What software was used for karyotype arrangement. How did authors distinguish homologous chromosome pairs? Did they use some chromosomal measurements?

4) Line 125: Include what tissue source authors used for cell culture establishment.

5) Line 126: Explain GTG and CBG abbreviations.

6) Lines 137-40: The nomenclature used e.g. in Session et al (2016) for polyploid Xenopus laevis was "XLA" followed by number of a chromosome and the letter "L" or "S". Letters L and S indicate subgenome of allotetraploid species (e.g., XLA 1L). There is no correlation or distant correlation of nomenclature used in proposed MS with study Session et al (2016). For second the primary source of X. laevis nomenclature is Matsuda et al (2015).

7) Line 141: Ohnologs or orthologs?

8) Starting on the line 141: For clearer explanation I would prefer chromosomal homology relationship (homeologous and orthologous only) as it used for diploid-tetraploid Xenopus by Song et al (in press) https://matthiasstoeckdotorg.files.wordpress.com/2020/08/bevans_accversion.pdf. (page 3, starting by the line 47). If authors of proposed MS would like to keep their chromosomal homology terms, could they assign their terms with terms used by Song et al (in press)? 

9) Line 152: "In total, 21 cells were counted for chromosomes." Could you formulate the sentence more clearly?

10) Line 152: "The number of elements..." Specify which elements. Chromosomal? banded? Are there any banding patterns?

11) Figure 2 and S4. It is beautiful picture of chromosomes and it is difficult to construct a karyotype with high number of chromosomes such as A. baerii (2n = 248). But still I can see some discrepancies. e.g., q arm of one of homologs 5 is shorter; one of homologs 7 is much smaller than the other; one of homologs 10 seems to be acrocentric and not submeta-/metacentric. No. 103 and 104 of the Figure 2 do not correspond to no. 103 and 104 of the Figure S4. Also, chromosomes within homologous pairs 103 and 104 can be reorganized. For these reasons I am going back to my question at point 3: Based on what did authors distinguish homologous chromosomes from each other?

12) All figures: Scale bars are missing.

13) Line 218: The paragraph starts with "this repeat" is it Arut 434A?

14) Figure 5: a, b, c, d letters are not in bold.

15) Lines 283-4: The sentence is doubled.

16) Table 1: I would use different title for column "2n", e.g. chromosome number (or put an additional clear explanation). There is 2n, 4n and 6n inside the column "ploidy level"; and there is 2n for all listed species in column "2n".

17) Table 1: Could you include snRNA marker into "Molecular markers" column?

18) Table 2: Explain abbreviation "cent".

19) Line 408: dediploidization or re-diploidization?

There is a lot of suggestions for MS update but I believe that the proposed MS will be accepted after revision and editor consideration.

References:

Kalous, L., & Knytl, M. (2011). Karyotype diversity of the offspring resulting from reproduction experiment between diploid male and triploid female of silver Prussian carp, Carassius gibelio (Cyprinidae, Actinopterygii). Folia Zoologica60(2), 115-121.

Matsuda, Y., Uno, Y., Kondo, M., Gilchrist, M. J., Zorn, A. M., Rokhsar, D. S., ... & Taira, M. (2015). A new nomenclature of Xenopus laevis chromosomes based on the phylogenetic relationship to Silurana/Xenopus tropicalis. Cytogenetic and genome research145(3-4), 187-191.

Session, A. M., Uno, Y., Kwon, T., Chapman, J. A., Toyoda, A., Takahashi, S., ... & Van Heeringen, S. J. (2016). Genome evolution in the allotetraploid frog Xenopus laevis. Nature538(7625), 336-343.

Song X-Y, Furman BLS, Premachandra T, Knytl M, Cauret CMS, Wasonga DV, Measey J, Dworkin I, Evans BJ (2020) Sex chromosome degeneration, turnover, and sex-biased expression of sex-linked transcripts in African clawed frogs (Xenopus)Philosophical Transactions of the Royal Society of London, B (in press).

Author Response

(The authors gave the same response as above.)

Round 2

Reviewer 4 Report

The proposed manuscript ms was improved but still nimor revision is needed.

1) Line 257: there is probably typo in explanation: "CBG (G-bands ...)".

2) Chromosome pairs 103 and 104 (Figure 2) do not correspond with same chromosome pairs (Figure S4). This has to be updated and all pairs should be checked again if each pair of Figure 2 corresponds with identical pair of Figure S4.

3) Figure 2 and Figure S4: Chromosomes within pair 10 are not homologous. Unambiguosly, one homolog is metacentric and second one is telocentric.

Author Response

Dear Reviewer,

Thank you for reviewing our manuscript and for your important remarks and comments on our paper. We considered all your comments and incorporated them into the manuscript text.
